# Reasoning Models Know When They're Right: Probing Hidden States for Self-Verification

**Anqi Zhang**[1], **Yulin Chen**[12], **Jane Pan**[1], **Chen Zhao**[12], **Aurojit Panda**[1], **Jinyang Li**[1], **He He**[1]
[1]New York University    [2]NYU Shanghai

## Abstract

Reasoning models have achieved remarkable performance on tasks like math and logical reasoning thanks to their ability to search during reasoning. However, they still suffer from *overthinking*, often performing unnecessary reasoning steps even after reaching the correct answer. This raises the question: *can models evaluate the correctness of their intermediate answers during reasoning?* In this work, we study whether reasoning models encode information about answer correctness through probing the model's hidden states. The resulting probe can verify intermediate answers with high accuracy and produces highly calibrated scores. Additionally, we find models' hidden states encode correctness of future answers, enabling early prediction of the correctness before the intermediate answer is fully formulated. We then use the probe as a verifier to decide whether to exit reasoning at intermediate answers during inference, reducing the number of inference tokens by 24% without compromising performance. These findings confirm that reasoning models do encode a notion of correctness yet fail to exploit it, revealing substantial untapped potential to enhance their efficiency.[1]

## 1 Introduction

Recent advances in reasoning models, such as OpenAI's o1 (OpenAI, 2024) and DeepSeek-R1 (DeepSeek-AI et al., 2025), have demonstrated significant progress in complex reasoning capabilities, particularly in domains such as mathematical problem solving (DeepMind, 2024; Zhou et al., 2023) and logical reasoning (Feng et al., 2023; Liu et al., 2025; Lam et al., 2024). A key advantage of reasoning models lies in their ability to search: they often explore multiple *reasoning paths* leading to different *intermediate answers* to the original problem before arriving at a final solution (Figure 1, left). While this search-based reasoning is beneficial, it also introduces inefficiencies. Previous studies (Chen et al., 2025; Sui et al., 2025) show that reasoning models tend to *overthink* by exploring additional reasoning paths even after reaching a correct answer.

This observation prompts the question: *to what extent can models evaluate the correctness of their intermediate answers during reasoning?* Existing works that interpret hidden states primarily focuses on non-reasoning models with short Chain-of-Thoughts (CoT) (Zhu et al., 2024; Xie et al., 2024; Li et al., 2024b; Kadavath et al., 2022; Orgad et al., 2024) , which cannot address our central research question of whether reasoning models know when they're right during their unique search behavior, especially given the observed overthinking phenomenon. The answer to this question is also crucial to preventing overthinking, either through a more targeted design of the training strategy or a better elicitation method.

We investigate this question by probing the model's hidden states for answer correctness. Specifically, we segment the long Chain-of-Thought (CoT) into chunks containing intermediate answers, and train a binary classifier to predict answer correctness from the model's hidden states at the answer positions (Figure 1).

---

Correspondence to: anqi.zhang@nyu.edu, hhe@nyu.edu

[1]The code and probes are available in: `https://github.com/AngelaZZZ-611/reasoning_models_probing`

Figure 1: An illustration of the probing method. On the left side, long CoT is parsed into multiple chunks, each corresponding to a reasoning path and contains an intermediate answer as termination. On the right side, representations for each chunk are obtained and probe is used to predict the probability of answer being correct.

We find that information about answer correctness is readily encoded in the model's internal representations. A simple probe can reliably extract this information, performing accurately on both in-distribution and out-of-distribution examples. Moreover, the probe is highly calibrated, with an expected calibration error (ECE) below 0.1. Our analysis also reveals that the model's hidden states contain "look-ahead" information: correctness can be predicted even before the intermediate answer is fully articulated. Notably, when applying the same probing method to traditional short CoT models, we observe a significant degradation in both accuracy and calibration performance, suggesting that the encoded correctness information is likely acquired during reasoning-specific training. We hypothesize that exposure to long CoT patterns with both correct and incorrect intermediate steps enables reasoning models to implicitly learn answer correctness, even without explicit supervised signals.

We also investigate whether reasoning models effectively use this information on answer correctness during inference. Because the trained probe is well-calibrated, we use the output score to measure the model's *confidence* in the current intermediate answer. Ideally, the model should reason at an optimal length if it is taking advantage of the well-encoded correctness information, i.e. it should stop reasoning when the confidence about an intermediate answer is high enough. We adopt the probe as a verifier and implement a confidence-based early-exit strategy by thresholding confidence scores from the probe. The strategy achieves up to 24% reduction in inference tokens without compromising accuracy. The improvement in efficiency with our verifier reveals that while reasoning models encode information about answer correctness, they do not efficiently use this internal knowledge during inference.

## 2 Related work

**Uncertainty estimation in LLMs.** Black-box techniques for estimating LLM uncertainty over their response have primarily focused on prompting the model to verbalize its confidence directly, often aggregating self-reported confidence scores over multiple samples (Lin et al., 2022; Tian et al., 2023). However, Xiong et al. (2024); Kapoor et al. (2024) find that white-box methods, including those that depend on internal model representations (Mielke et al., 2022), tend to perform better than black-box methods on confidence estimation. For instance, Azaria & Mitchell (2023); Burns et al. (2024) show that an LLM's representation after processing a statement is highly predictive of the statement's correctness; moreover, linear probes trained on these representations can classify correctness, even without ground-truth labels. We extend this work to long CoT generated by reasoning models, demonstrating that

the representations at intermediate stages of the CoT also capture key information about the correctness of each intermediate stage. Especially, we show reasoning models exhibit substantially better calibration, which is inline with more recent work (Yoon et al. (2025)) showing that reasoning models better express their confidence.

**Efficient reasoning during inference.**   Reasoning models demonstrate improved performance on many tasks thanks to their ability to search while generating reasoning chains, which often demand additional test-time compute in comparison to standard CoT (DeepSeek-AI et al., 2025). Additionally, reasoning models often suffer from repeated and unnecessary reasoning steps—or "overthinking"—even after a correct answer has been reached (Chen et al., 2025). Recent work has explored training methods to make reasoning more concise or to reduce the frequency of overthinking (Chen et al., 2025; Munkhbat et al., 2025). Other inference-time techniques focus on curtailing generations that are unlikely to be successful (Zhao et al., 2025; Manvi et al., 2024; Li et al., 2024a) or dynamically adjusting the test-time compute budget based on input difficulty or other properties of the prompt (Damani et al., 2024; Wang et al., 2025; Xu et al., 2024; Fu et al., 2024). We find that while the model encodes information about answer correctness, it fails to use it efficiently, which may contribute to overthinking. We leverage this to perform threshold-based early-exiting at inference time, reducing test-time compute while preserving performance.

**Learned verifiers.**   The ability to verify intermediate answers is also related to the line of works on verifiers, which is an important technique used to regulate test-time scaling. Previous work has focused on training verifiers to classify the correctness of a model-generated solution or select which of two model-generated responses is preferred (Bai et al., 2022; Cobbe et al., 2021; Lightman et al., 2023; Zhang et al., 2025; Zheng et al., 2023; Creswell & Shanahan, 2022; Paul et al., 2024). However, recent improvements in reasoning capabilities have enabled LLMs to critique and refine their outputs without the aid of external verifiers, often using natural language prompt templates to guide self-critique of model-generated output (Ling et al., 2023; Zhang et al., 2024; Madaan et al., 2023; Weng et al., 2023; Shinn et al., 2023). In contrast, we focus on leveraging information about correctness which is encoded in the model representations of the reasoning chain.

## 3   Probing for intermediate answer correctness

The long CoT output from a reasoning model often contains multiple mentions of *intermediate answers*. We aim to explore whether the notion of "correctness" is encoded in the representation of each intermediate answer by probing. This section describes how we identify intermediate answers, obtain their representations, and train a two-layer multilayer perceptron (MLP) probe.

### 3.1   Data collection

We first collect responses from reasoning models for each problem in the task dataset. The reasoning trace, which is encapsulated in *<think>* tokens, is extracted and split into paragraphs with "\n\n" as delimiter. We identify the start of a new reasoning path by detecting keywords like "wait", "double-check" and "alternatively" in each paragraph. A complete list of the keywords is shown in Table 3 in the appendix. We merge paragraphs in the same reasoning path to form a *chunk*. Then we use Gemini 2.0 Flash (Gemini-Team, 2024) to extract the intermediate answer in each chunk if one exists, and judge its correctness against the true answer. Finally, adjacent chunks that do not contain an intermediate answer are merged with the nearest chunk that contains an answer. Each merged chunk now has an intermediate answer and a label generated by Gemini, represented as $\{(c_1, y_1), (c_2, y_2), ...(c_k, y_k)\}$, where each $c_i$ is part of the reasoning trace that contains an answer to the original problem, and $y_i$ is a binary label indicating the correctness of the answer. § A.3 presents results on the sensitivity of LLM-based annotation with different advanced LLMs. The high consistency shows that this task is relatively straightforward for state-of-the-art LLMs.

The next step is to obtain the model representation for each chunk. For each chunk $c_i$, we take the last-layer hidden states at the last token position as its representation $e_i$. Finally, for each task dataset, we collect a set of reasoning representations and their corresponding labels, formulating the probing dataset $\mathcal{D} = \{(e_i, y_i)\}_{i=1}^{N}$ that will be finally used to train probes. Note that the construction of probing dataset $\mathcal{D}$ depends on both the original task dataset and the reasoning model we use to generate representations.

## 3.2 Training the probe

After obtaining the probing dataset, we train a two-layer multilayer perceptron on $\mathcal{D}$. Since the datasets are often highly imbalanced, where most intermediate answers from a strong reasoning models are correct (see Table 4 in Appendix A.1 for detailed label statistics), we use weighted binary cross-entropy loss:

$$p_i = \sigma(\text{ReLU}(e_i \mathbf{W}_1 + \mathbf{b}_1)\mathbf{W}_2 + b_2)$$
$$\mathcal{L}(\mathbf{W}, \mathbf{b}) = -\frac{1}{N}\sum_{i=1}^{N}\left(w\alpha y_i \log p_i + (1 - y_i)\log(1 - p_i)\right) \tag{1}$$

where $\sigma$ is the sigmoid function, $w$ is the ratio of negative to positive samples in the training data, and $\alpha$ is a hyperparameter to scale the imbalance weight. The model parameters are $\mathbf{W}_1 \in \mathbb{R}^{m \times d}$, $\mathbf{W}_2 \in \mathbb{R}^{d \times 1}$, $\mathbf{b}_1 \in \mathbb{R}^d$, and $b_2 \in \mathbb{R}$, where $m$ is the hidden size of the language model and $d$ is the hidden size of the MLP.

# 4 Experiments

We first describe the basic experimental setup (§ 4.1). Then, we explore whether information about answer correctness is encoded in reasoning models (§ 4.2) and if it generalizes across datasets (§ 4.3), how such information is related to long CoT reasoning abilities (§ 4.4), and is the information also well-encoded even before an explicit answer is formulated (§ 4.5).

## 4.1 Experimental setup

**Task datasets.** We select mathematical reasoning and logical reasoning tasks as their answers are automatically verifiable. For mathematical reasoning, we use three datasets: GSM8K (Cobbe et al., 2021), MATH (Hendrycks et al., 2021), and AIME. For logical reasoning, we use KnowLogic (Zhan et al., 2025), a logical reasoning benchmark of 5.4k multiple-choice questions synthesized with knowledge-driven methods. For expert-level scientific QA, we use GPQA (Rein et al., 2023). To ensure the reliability of intermediate answer extraction, we filter the KnowLogic dataset to only retain examples with a single correct answer. For ease of training, all training sets are down-sampled to include no more than 1000 examples, which did not affect performance according to our pilot experiment. See Appendix A.1 for more details regarding data processing.

**Reasoning models.** We use the open-source DeepSeek-R1-Distill series of models (DeepSeek-AI et al., 2025), including R1-Distill-Llama-8B, R1-Distill-Llama-70B, R1-Distill-Qwen-1.5B, R1-Distill-Qwen-7B, and R1-Distill-Qwen-32B. All the distilled models are supervised fine-tuned with reasoning data generated by DeepSeek-R1 model. We also use QwQ-32B (Team, 2025; Yang et al., 2024), an open-source reasoning language model trained with reinforcement learning.

**Implementation details.** For probing data collection, we enumerate each combination of task dataset and model to collect model representation and answer labels. The statistics of the collected data can be found in Appendix A.1. For training, each dataset $\mathcal{D}$ is randomly split into a training set and a validation set $\mathcal{D}_{train}$ and $\mathcal{D}_{val}$, with a train-to-validation ratio of 8:2. The Adam optimizer (Kingma & Ba, 2017) is used for training, and we perform grid search for hyperparameter tuning. The hyperparameters for search include learning rate, scaling factor for imbalance weight $\alpha$, weight decay, and MLP hidden size $d$. Each

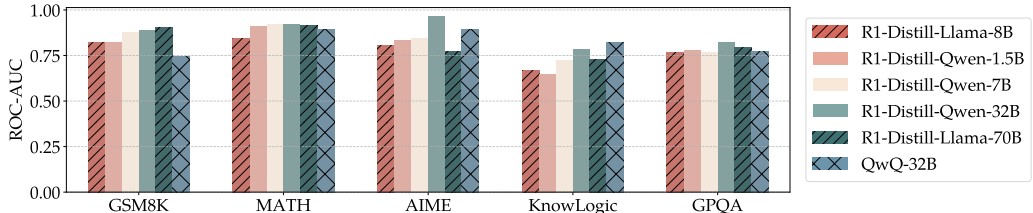

Figure 2: ROC-AUC scores for each probe trained on hidden states from different reasoning models and datasets. We train a separate probe on each probing dataset and evaluate it on in-distribution test set.

model is trained for at most 200 epochs with a batch size of 64; the validation loss is used as the criterion for early stopping. Following grid search, the probing models are first ranked based on their validation accuracy. From the top 10 performing models, we select the probe with the least number of parameters, specifically the model with the smallest hidden dimension $d$. Details regarding the grid search setting and search results for each probing dataset can be found in Appendix A.4. Note that most resulting models achieve non-trivial performance when $d = 0$ (see Appendix A.4), which means that correctness of the intermediate answer can be easily extracted with a linear probe.

## 4.2 Reasoning models encode answer correctness

We first test **in-distribution** performance of trained probes by evaluating each probe on the test set from the same dataset as the training set. Figure 2 reports the ROC-AUC scores on each dataset, and Table 1 presents the corresponding Expected Calibration Error (ECE) (Naeini et al., 2015) and Brier score (Brier, 1950). Other metrics including accuracy, precision, recall, and macro F1 are reported in Appendix A.6.

Overall, all probes perform satisfactorily in in-distribution setting, achieving ROC-AUC scores above 0.7 and remarkably low Expected Calibration Error (ECE) scores below 0.1. This indicates the reasoning models inherently encode information about answer correctness that can be extracted with a simple probe. Moreover, many of the probes converge to a linear probe after grid search (hidden size $d = 0$), suggesting that correctness information is linearly encoded in the hidden states of the reasoning model (Appendix A.4).

Across task datasets, probes trained on mathematical reasoning data perform better than those trained on logical reasoning data. This may correlate with the training data distributions of the reasoning models, where math problems presumably play a larger role. Meanwhile, probes extracted from larger reasoning models work better, with R1-Distill-Qwen-32B achieving over 0.9 ROC-AUC score on AIME. The Qwen family models' representations also exhibit stronger correctness signals, with Qwen-1.5B generally surpassing Llama-8B model in the mathematical domain, potentially reflecting differences in the base model training data distribution.

| Reasoning Model | GSM8K | | MATH | | AIME | | KnowLogic | | GPQA | |
|---|---|---|---|---|---|---|---|---|---|---|
| | ECE ↓ | Brier ↓ | ECE ↓ | Brier ↓ | ECE ↓ | Brier ↓ | ECE ↓ | Brier ↓ | ECE ↓ | Brier ↓ |
| R1-Distill-Llama-8B | 0.05 | 0.17 | 0.03 | 0.14 | 0.10 | 0.11 | 0.07 | 0.23 | 0.09 | 0.19 |
| R1-Distill-Llama-70B | 0.03 | 0.07 | 0.07 | 0.10 | 0.10 | 0.18 | 0.03 | 0.19 | 0.03 | 0.18 |
| R1-Distill-Qwen-1.5B | 0.04 | 0.16 | 0.04 | 0.12 | 0.14 | 0.12 | 0.09 | 0.20 | 0.05 | 0.14 |
| R1-Distill-Qwen-7B | 0.02 | 0.11 | 0.03 | 0.10 | 0.09 | 0.15 | 0.06 | 0.21 | 0.05 | 0.18 |
| R1-Distill-Qwen-32B | 0.01 | 0.08 | 0.06 | 0.09 | 0.13 | 0.10 | 0.10 | 0.19 | 0.09 | 0.18 |
| QwQ-32B | 0.03 | 0.13 | 0.13 | 0.10 | 0.08 | 0.13 | 0.03 | 0.15 | 0.09 | 0.20 |

Table 1: Expected Calibration Error (ECE) and Brier score for the in-distribution performance of each probe trained on each probing dataset.

### 4.3 Probes generalize to some out-of-distribution datasets

Past studies have shown that probe performance can deteriorate significantly when applied to out-of-distribution data (Belinkov, 2021; Kapoor et al., 2024). Since strong in-distribution results may not necessarily indicate reliable generalization, we examine how well the probes trained in § 4.2 perform across different domains and datasets.

Table 2 shows the ROC-AUC and ECE scores for probes evaluated on out-of-distribution data, compared to those trained and tested on in-distribution data, using representations from R1-Distill-Llama-8B. We find that probes exhibit generalizability across mathematical reasoning datasets. The probes trained on MATH and GSM8K transfer well between the two datasets, demonstrating both high discriminative performance (ROC-AUC) and satisfactory calibration (ECE). In contrast, for AIME, a more difficult dataset, the probes trained on GSM8K and MATH are less calibrated. However, the probe does not stably generalize to out-of-domain data (e.g., from logical reasoning to mathematical reasoning), perhaps due to the difference in distribution of the two domains (Figure 6). More generalization results on other reasoning models can be found in Appendix A.6.

| Training Data | GSM8K | | MATH | | AIME | | KnowLogic | |
|---|---|---|---|---|---|---|---|---|
| | AUC ↑ | ECE ↓ | AUC ↑ | ECE ↓ | AUC ↑ | ECE ↓ | AUC ↑ | ECE ↓ |
| GSM8K | 0.82 | 0.05 | 0.80 (-0.04) | 0.08 (+0.05) | 0.69 (-0.11) | 0.25 (+0.15) | 0.56 (-0.11) | 0.10 (+0.03) |
| MATH | 0.83 (+0.01) | 0.04 (-0.01) | 0.84 | 0.03 | 0.76 (-0.04) | 0.28 (+0.18) | 0.63 (-0.04) | 0.08 (+0.01) |
| KnowLogic | 0.77 (-0.05) | 0.17 (+0.12) | 0.74 (-0.10) | 0.19 (+0.16) | 0.81 (+0.01) | 0.31 (+0.21) | 0.67 | 0.07 |

Table 2: ROC-AUC scores and ECE of trained probes on out-of-distribution test set. The numbers in red and green denote performance decrease and increase relative to the probe trained on in-distribution training set, respectively. R1-Distill-Llama-8B is used as the reasoning model.

### 4.4 Encoding of correctness is related to long CoT reasoning abilities

We have shown information on answer correctness is encoded in reasoning model's hidden states; to what extent this encoding is related to the model's ability to perform long CoT reasoning? To that end, we train a probe with the non-reasoning counterpart of the reasoning model. Specifically, we use Llama-3.1-8B-Instruct (Grattafiori & Others, 2024) to obtain representations of reasoning chunks using the MATH dataset. As instruct models do not have long CoT reasoning abilities, each chunk is just the full model output for one problem (i.e., including the short CoT and final answer), and the representation is simply the hidden state of the last token for each problem output. To account for this, we add an additional setting for reasoning model probes, where the probe is evaluated on the correctness of the final answers (rather than the intermediate answers) of each reasoning chain.

As shown in Figure 3, the probe trained on non-reasoning model representations performs much worse than its reasoning counterpart, with lower classification scores and higher calibration errors. The fact that the encoded information on answer correctness is more prominent in reasoning models may suggest that the self-verification ability is enhanced during long CoT supervised training. The enhanced correctness encoding might have to do with training data difference. Non-reasoning models are trained with question-response pairs that contain a single correct answer, whereas reasoning models usually undergo long CoT supervision which contains both correct and incorrect intermediate answers, often with self-correction or backtracking behaviors following the incorrect ones. This exposure may enable reasoning models to implicitly learn the correctness of intermediate answers (which is useful in predicting the reasoning chain), even without explicit supervised signals.

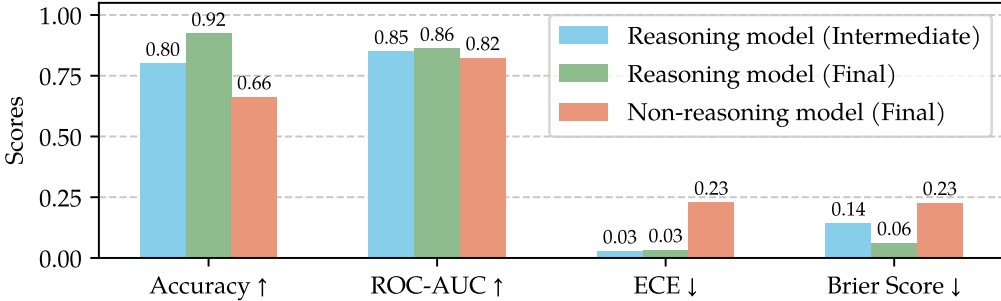

Figure 3: Comparison on the performance on reasoning models (i.e., R1-Distill-Llama-8B, fine-tuned on the base Llama-3.1-8B model using long CoT data) and non-reasoning models (i.e., Llama-3.1-8B-Instruct) on MATH. For reasoning models, we show both the performance on predicting the correctness of intermediate answers (blue) and the final answers (green). For non-reasoning models, the data only contains the final answers (red).

### 4.5 Correctness can be detected before the answer is generated

Section 4.2 shows that the hidden states at the *end* of reasoning chunks encode information about intermediate answer correctness, we now investigate a further question: do hidden states from earlier positions within the chunk also encode such signals? Specifically, we analyze hidden states from varying positions *midway* through a reasoning chunk—before an intermediate answer is fully generated—to determine if these earlier representations already encode predictive signals about the forthcoming answer's correctness.

As described in § 3.1, each reasoning trace is initially split into $k$ chunks with corresponding correctness labels $\{(c_1, y_1), (c_2, y_2), ...(c_k, y_k)\}$. Each chunk $c_i$ can be subdivided into paragraphs. We obtain the representation of each paragraph-level sequence, and assign each sequence within chunk $c_i$ the label $y_i$, corresponding to the correctness of the nearest upcoming intermediate answer. We train a probe to predict the future answer correctness for R1-Distill-Llama-8B on MATH (following § 3.2). We use hidden states at the end of different paragraphs to predict chunk correctness. We report probing performance at different percentages of all paragraphs within a chunk.

We observe that the reasoning model's hidden states encode information about correctness even before an intermediate answer has been explicitly generated. Moreover, the probe performance is positively correlated with the paragraph's proximity to the upcoming intermediate answer. As shown in Figure 4, the probe's classification accuracy improves primarily during two critical phases: an initial steep increase in the 0-10% range, followed by minimal gains until a second noticeable improvement near the chunk's end (90-100%). Compared to the peak accuracy of 79%, performance at the 10%, 50%, and 95% positions shows decrements of 14%, 10%, and 5% respectively. This highlights that early positions contain significant correctness signals, while the most predictive information emerges just before answer generation. On the other hand, calibration error is highest at the initial paragraph and then undergoes a sharp decline. ECE reaches its minimum (0.03) relatively early——at around the 60% position——while the Brier score continues improving until the final positions of the reasoning chunk.

## 5 Probe as a verifier for early-exit

While reasoning models are able to encode well-calibrated and accurate information about intermediate answer correctness, do they fully utilize it during inference? We investigate this by checking whether early exiting based on the probe's confidence score on answer correctness can improve reasoning efficiency. This approach allows us to determine whether models continue reasoning unnecessarily after the probe is highly confident that the answer is correct (i.e., overthinking).

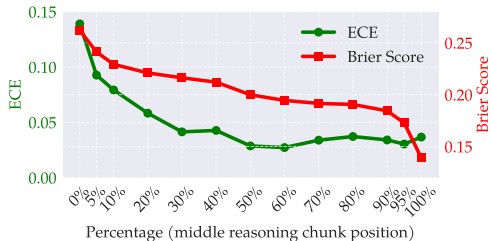 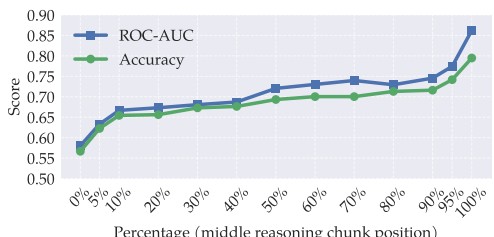

(a) ECE and Brier Score decrease as the paragraph position approaches the answer at the end of the reasoning chunk

(b) Accuracy and ROC-AUC increase as the paragraph position approaches the generated answer at the end of the reasoning chunk

Figure 4: Performance on predicting the correctness of the upcoming intermediate answers midway through a reasoning chunk. The results are obtained at different percentages of all paragraphs within each chunk. The task dataset and reasoning model used are MATH dataset and R1-Distill-Llama-8B.

## 5.1 Experimental setup

Following § 3, we obtain a classifier that takes a reasoning chunk $c_i$'s representation $e_i$ as input and outputs the probability $p_i$ of the intermediate answer $y_i$ being correct. Since the estimated $p_i$ is highly calibrated (§ 4.2), we directly use it to guide **confidence-based early-exit** during inference. Specifically, we first set a threshold *Thr* for model confidence. Then, we sequentially evaluate each intermediate answer in the full reasoning trace, using the probe to compute confidence scores on the answer's correctness. Once we encounter an intermediate answer whose probed $p_i$ exceeds the threshold *Thr*, we truncate the reasoning trace at this chunk and take the intermediate answer as the final answer.

We compare the intermediate answer selected by early exiting with the question's ground-truth answer to compute accuracy. Additionally, we record the inference token length at the point of truncation to evaluate computational efficiency. We run R1-Distill-Llama-8B on MATH dataset. In this experiment, the maximum token generation limit is set to be 10K across all test examples.

For comparison, we implement **static early-exit**, where we predetermine a fixed number of intermediate answers $m$ and terminate the reasoning process after $m$ chunks, taking the $m$-th chunk's intermediate answer $y_m$ as the final answer[2].

We also compare the probed confidence score with verbalized confidence (details in § A.5), where we prompt the same reasoning model to output its evaluation of the current answer for each chunk.

## 5.2 Results

As shown in Figure 5, using the probe to perform confidence-based early exiting can improve reasoning efficiency without accuracy degradation. When setting *Thr* to 0.85, our strategy achieves roughly the same reasoning accuracy (88.2%) as no early-exit, while reducing the number of generated tokens by approximately 24%. Setting *Thr* to 0.9 (or higher) can achieve identical reasoning accuracy (88.6%) as no early-exit and reduces the number of generated tokens by 19%. In other words, without early exiting, the reasoning model continues to generate excess tokens even when the probe indicates high confidence; this failure to fully utilize internal information on answer correctness empirically leads to overthinking behavior.

Additionally, when saving equivalent numbers of tokens, our approach outperforms the static early-exit strategy by achieving up to a 5% accuracy improvement. For instance,

---

[2]Note that the static early-exit strategy degrades to no early-exit if the total number of chunks $k < m$.

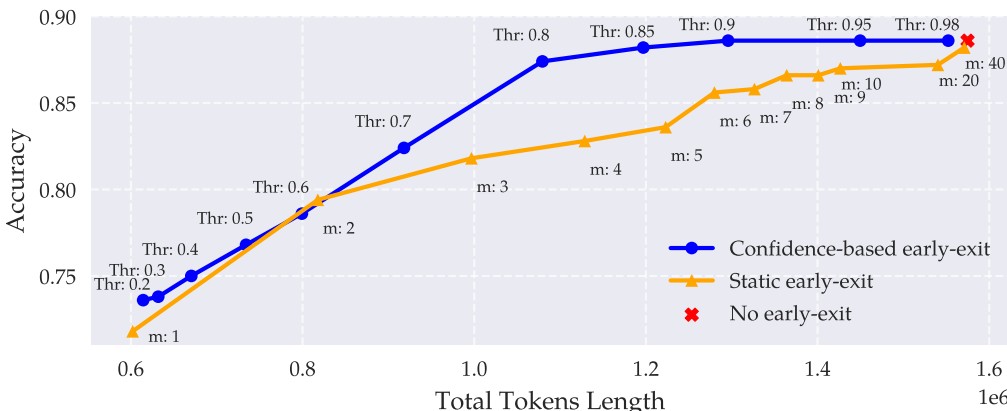

Figure 5: Final answer accuracy versus inference token cost with different early-exit strategies. For confidence-based early-exit, the curve is obtained by varying the confidence threshold for answer correctness. For static early-exit, the curve is generated by varying the chunk number $m$.

confidence-based early exiting has 87.4% accuracy ($Thr = 0.8$), whereas the static early-exit strategy has approximately 82.5% accuracy with similar total token usage. Controlling for the same accuracy score (e.g. above 85%), confidence-based early-exit strategy ($Thr = 0.8$) consumes significantly fewer tokens than static strategy (with $m = 6$). This demonstrates that leveraging the internal encoded information of answer correctness as an exit strategy can lead to more efficient reasoning.

Figure 7 and Figure 8 in § A.5 show that the probed confidence outperforms the verbalized confidence score in terms of both accuracy and token efficiency.

Overall, the improvements suggest that reasoning models fail to fully leverage this internal encoded information of answer correctness during inference, and that more effective usage of the information can reduce overthinking and enhance reasoning efficiency.

## 6 Discussion

In this study, we explore the existence of answer correctness information in reasoning models' inner representation. With probing, we show that such information is readily accessible in models' hidden states. The trained probe demonstrate strong calibration performance, and can be adopted as a lightweight verifier to improve reasoning efficiency. The significant reduction in inference tokens suggest that reasoning models' hidden states probably contain rich information that are underexplored. Our findings contribute to the growing body of research on model interpretability and open up several intriguing avenues for future investigation.

**Self-verification ability of language models.** Our study reveals that answer correctness is encoded in reasoning models' hidden states. The information can be easily extracted with a probe and used as a verifier during inference. This indicates that strong self-verification abilities can be elicited from reasoning models. Notably, these abilities are less pronounced in non-reasoning models. However, given the intricate training processes and the diversity of training data these models are exposed to, the precise origins of this ability remains unclear, suggesting a promising avenue for future research into how and when such self-verification abilities emerge during model training.

**Internal mechanisms of reasoning models.** We uncover a surprisingly well-calibrated hidden verifier that enables models to autonomously assess intermediate reasoning correctness. This finding suggests that models possess an ability to self-verify, which is an important step toward understanding their internal decision-making processes. However, we still

observe "overthinking" phenomenon, where models perform unnecessary re-checks even after generating correct answers with high confidence, as demonstrated in our early-exit experiment. This suggests that while models can self-verify, they do not yet efficiently leverage this intrinsic capability. Further study is needed to explore how reasoning models internally utilize the information encoded in their representations, and how we can guide them to use this information more efficiently during training or inference.

**On-policy control of reasoning models.** In contrast to previous LLM-based verifiers (Zhang et al., 2025; Cobbe et al., 2021; Zhang et al., 2024), the hidden verifier extracted in our work is much more lightweight. Our approach leverages the hidden states of the reasoning model directly during inference, which not only improves token efficiency but also makes the verifier more integrated with the model's existing architecture. Our finding highlights the potential of an on-policy perspective in model inference control. We believe this opens new avenues for future research in designing more efficient and adaptive control modules for reasoning models.

In summary, our study highlights the encoded answer correctness information in reasoning models, indicating the latent capability of reasoning models to verify their own answers. Leveraging this information through lightweight probing techniques, we show reasoning efficiency can be further enhanced, implying an inadequate use of the information by reasoning models during inference. Our findings underscore the potential of on-policy control for reasoning models, offering a novel direction for more efficient and adaptive inference strategies. Future research should further investigate the origins of the self-verification abilities and develop methods to better harness them, ultimately improving reasoning efficiency and reliability.

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

# A  Additional details

## A.1  Data collection details

Table 3 shows the keywords we use to identify beginning of new reasoning paths to help segmenting reasoning trace into chunks.

For AIME, we use AIME_1983_2024 [3] for training and AIME_2025 [4] for testing. For MATH, we use the original training set and the 500-example test set released by HuggingFace [5]. For KnowLogic dataset, we randomly split the dataset into a training and

| Keywords for chunk segmentation |
| --- |
| "wait", "double-check", "alternatively", "make sure", "another way", "verify", "to confirm" |

Table 3: Keywords we use for identifying reasoning path switch and segmenting reasoning trace into chunks.

test set by 80% and 20%, and collect probing data separately. For GPQA dataset [6], we use the GPQA_diamond subset as test data and the rest of GPQA_extended as training data.

Table 4 shows the statistics of collected chunks for each dataset. We use vLLM (Kwon et al., 2023) for inference and set maximum output length to 30K. Examples whose model completion goes over the maximum output length are discarded.

Figure 6 is a visualization of chunk representations obtained for different datasets with R1-Distill-Llama-8B (DeepSeek-AI et al., 2025). The domain difference between logical reasoning and mathematical problems is evident.

## A.2  Prompts

Table 5 shows the prompt we used to elicit reasoning trace from all reasoning models. Note that for Qwen models, the prompt we use is slightly different from its original prompt. We observe the performance on the benchmark does degrade a little but within a reasonable range. To ensure the extracted feature is on-policy, we also keep the same prompt when extracting representations for each reasoning chunk.

Table 6 is the evaluation prompt we use for Gemini 2.0 Flash (Gemini-Team, 2024) for answer extraction and evaluation based on given reasoning chunks.

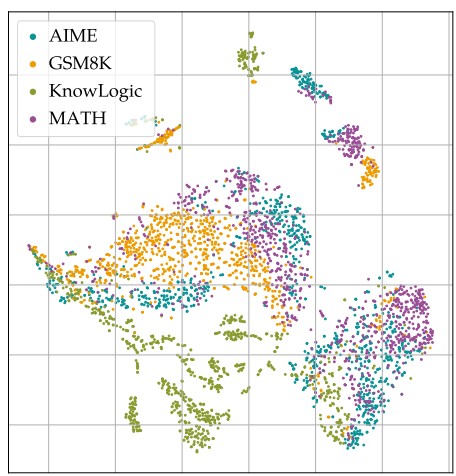

Figure 6: T-SNE visualization of chunk representations for different datasets. 1000 chunks are randomly sampled from each training set and R1-Distill-Llama-8B is used to obtain the representation.

---

[3]https://huggingface.co/datasets/di-zhang-fdu/AIME_1983_2024
[4]https://huggingface.co/datasets/yentinglin/aime_2025
[5]https://huggingface.co/datasets/HuggingFaceH4/MATH-500
[6]https://huggingface.co/datasets/Idavidrein/gpqa

| Reasoning Model | #Train Examples | #Test Examples | #Train Chunks | #Test Chunks | Avg. Chunk Len. | Positive Chunks (%) |
|---|---|---|---|---|---|---|
| | | | *GSM8K* | | | |
| R1-Distill-Llama-8B | 1000 | 1317 | 7379 | 11228 | 328.0 | 70.97 |
| R1-Distill-Llama-70B | 998 | 1318 | 9030 | 6116 | 272.4 | 84.36 |
| R1-Distill-Qwen-1.5B | 995 | 1308 | 8599 | 11730 | 379.0 | 63.57 |
| R1-Distill-Qwen-7B | 1000 | 1316 | 5615 | 7568 | 302.8 | 75.87 |
| R1-Distill-Qwen-32B | 996 | 1317 | 4393 | 6381 | 293.1 | 84.25 |
| QwQ-32B | 1000 | 1319 | 4786 | 6302 | 338.2 | 82.78 |
| | | | *MATH* | | | |
| R1-Distill-Llama-8B | 1000 | 491 | 6259 | 3380 | 615.1 | 76.91 |
| R1-Distill-Llama-70B | 996 | 499 | 4865 | 2559 | 701.5 | 82.88 |
| R1-Distill-Qwen-1.5B | 988 | 495 | 7388 | 4089 | 996.7 | 68.46 |
| R1-Distill-Qwen-7B | 983 | 494 | 5062 | 2764 | 713.3 | 79.77 |
| R1-Distill-Qwen-32B | 991 | 497 | 4732 | 2460 | 678.5 | 84.40 |
| QwQ-32B | 1000 | 500 | 5619 | 2826 | 795.0 | 86.12 |
| | | | *AIME* | | | |
| R1-Distill-Llama-8B | 922 | 30 | 7158 | 323 | 1652.0 | 35.24 |
| R1-Distill-Llama-70B | 923 | 30 | 5443 | 318 | 1528.0 | 50.78 |
| R1-Distill-Qwen-1.5B | 892 | 29 | 8358 | 314 | 1809.4 | 26.33 |
| R1-Distill-Qwen-7B | 922 | 29 | 5501 | 179 | 1841.8 | 42.50 |
| R1-Distill-Qwen-32B | 868 | 25 | 4181 | 104 | 1244.1 | 55.03 |
| QwQ-32B | 933 | 30 | 5037 | 142 | 2079.1 | 68.14 |
| | | | *KnowLogic* | | | |
| R1-Distill-Llama-8B | 986 | 320 | 7620 | 2596 | 1079.6 | 44.27 |
| R1-Distill-Llama-70B | 996 | 297 | 6529 | 2000 | 639.7 | 57.71 |
| R1-Distill-Qwen-1.5B | 762 | 245 | 6879 | 2036 | 1070.0 | 20.56 |
| R1-Distill-Qwen-7B | 938 | 306 | 7169 | 2430 | 1072.7 | 42.25 |
| R1-Distill-Qwen-32B | 979 | 315 | 6131 | 1827 | 818.8 | 57.40 |
| QwQ-32B | 1000 | 326 | 6256 | 2093 | 1043.3 | 70.78 |
| | | | *GPQA* | | | |
| R1-Distill-Llama-8B | 348 | 198 | 2891 | 1662 | 794.6 | 38.89 |
| R1-Distill-Llama-70B | 348 | 198 | 1477 | 1151 | 737.5 | 51.46 |
| R1-Distill-Qwen-1.5B | 348 | 198 | 2618 | 1745 | 803.9 | 17.61 |
| R1-Distill-Qwen-7B | 348 | 198 | 2547 | 1601 | 852.2 | 33.73 |
| R1-Distill-Qwen-32B | 348 | 198 | 1796 | 1188 | 747.0 | 41.76 |
| QwQ-32B | 348 | 198 | 1477 | 1220 | 1067.7 | 45.09 |

Table 4: Statistics for obtained probing dataset across task datasets and reasoning models. The inconsistency in training examples and test examples number comes from discard of examples with truncated model completion. The average chunk length is calculated by sampling 1000 chunks from each training dataset and measured by number of tokens. The positive chunk ratio is calculated based on the training set.

| **Inference Prompt** |
|---|
| $<$BOS_TOKEN$> <$|User|$>$ {instruction} |
| Please reason step by step, and put your final answer within \boxed{}. |
| $<$|Assistant|$>$ |

Table 5: Prompt used for inference with reasoning models.

## A.3 Sensitivity of LLM annotation

To test the sensitivity of LLM annotation, we compare annotation results from Gemini with GPT-4o-mini with randomly sampled 100 examples from MATH training data. As shown in the Table. 7 below, the two models well agree with each other with 80% annotations

---

**Evaluation Prompt**

---

Given several chunks of a reasoning trace, along with a ground-truth answer, independently evaluate each chunk. If a chunk reaches a result at the end, return the intermediate result; otherwise, return None if the chunk does not contain an intermediate result (e.g., pure reflections).

Then, if an intermediate answer exists, compare it to the ground-truth answer. If the intermediate result in the chunk equals the ground-truth answer, return True; if the intermediate result in the chunk does not equal the ground-truth answer, return False; if no intermediate answer exists, return None.

Output in JSON format:

```
[
  {"id": "1", "result": "6 + 9i" / None, "correctness": True / False /
None},
  …
]
```

Input chunks: {reasoning_trace}
Ground-truth answer: {answer}

---

Table 6: Prompt used for answer extraction and evaluation with Gemini 2.0 Flash.

exactly the same. The results show that the annotation task is relatively straightforward for state-of-the-art LLMs.

| CoT Data | Cohen's kappa coefficient | % consistent annotation | # segments |
|---|---|---|---|
| Qwen1.5B+MATH | 0.58 | 85 | 3416 |
| Qwen7B+MATH | 0.62 | 80 | 1772 |
| Qwen32B+MATH | 0.60 | 81 | 1513 |

Table 7: Annotation consistency and agreement for Gemini and GPT-4o-mini models.

## A.4 Grid search

We perform grid search over hyparameters include learning rate, loss weight scaling factor $\alpha$, weight decay for optimizer, and classifier hidden size $d$. The specific search range for each hyperparameter can be found in Table 8, and the resulting optimal hyperparameter settings for each probing dataset are shown in Table 9.

| Hyperparameter | Search Space |
|---|---|
| Learning rate | 1e-3, 1e-4, 1e-5 |
| Scaling factor $\alpha$ | 0.3, 0.5, 0.7, 0.9, 1.0, 1.5, 2.0, 3.0 |
| Weight decay | 0.001, 0.01, 0.1 |
| Hidden size $d$ | 0, 16, 32 |

Table 8: Hyperparameter search space for classifier training.

| Model | Dataset | Learning rate | Loss weight $\alpha$ | Weight decay | Hidden size $d$ |
|---|---|---|---|---|---|
| R1-Distill -Llama-8B | GSM8K | 1e-4 | 3.0 | 0.1 | 16 |
|  | MATH | 1e-5 | 2.0 | 0.001 | 0 |
|  | AIME | 1e-5 | 0.3 | 0.1 | 0 |
|  | KnowLogic | 1e-5 | 0.7 | 0.1 | 0 |
|  | GPQA | 1e-5 | 0.3 | 0.001 | 0 |
| R1-Distill -Qwen-1.5B | GSM8K | 1e-5 | 2.0 | 0.1 | 16 |
|  | MATH | 1e-3 | 2.0 | 0.01 | 16 |
|  | AIME | 1e-5 | 0.5 | 0.01 | 16 |
|  | KnowLogic | 1e-4 | 0.3 | 0.001 | 0 |
|  | GPQA | 1e-5 | 0.3 | 0.001 | 0 |
| R1-Distill -Qwen-7B | GSM8K | 1e-4 | 3.0 | 0.1 | 0 |
|  | MATH | 1e-4 | 3.0 | 0.1 | 0 |
|  | AIME | 1e-3 | 0.9 | 0.1 | 0 |
|  | KnowLogic | 1e-5 | 0.9 | 0.1 | 0 |
|  | GPQA | 1e-5 | 0.5 | 0.01 | 0 |
| R1-Distill -Qwen-32B | GSM8K | 1e-3 | 3.0 | 0.001 | 16 |
|  | MATH | 1e-4 | 2.0 | 0.1 | 0 |
|  | AIME | 1e-5 | 1.0 | 0.01 | 16 |
|  | KnowLogic | 1e-5 | 0.9 | 0.1 | 0 |
|  | GPQA | 1e-5 | 1.5 | 0.01 | 0 |
| R1-Distill -Llama-70B | GSM8K | 1e-4 | 2.0 | 0.001 | 0 |
|  | MATH | 1e-4 | 3.0 | 0.001 | 0 |
|  | AIME | 1e-4 | 2.0 | 0.001 | 0 |
|  | KnowLogic | 1e-3 | 1.0 | 0.01 | 32 |
|  | GPQA | 1e-4 | 0.9 | 0.001 | 0 |
| QwQ-32B | GSM8K | 1e-4 | 3.0 | 0.1 | 0 |
|  | MATH | 1e-3 | 2.0 | 0.001 | 16 |
|  | AIME | 1e-3 | 3.0 | 0.01 | 16 |
|  | KnowLogic | 1e-4 | 1.5 | 0.1 | 0 |
|  | GPQA | 1e-5 | 0.7 | 0.01 | 0 |

Table 9: Results of grid search across reasoning models and datasets.

## A.5 More results about prompting for confidence verbalization

Prompting the model to verbalize its confidence is a straightforward method to test its notion of correctness. However, in real implementation, we find it extremely difficult to get reasoning models to follow user instruction. Experimenting with different prompting strategies, we only manage to get the model to produce a verbalized confidence for 66% of the chunks. On these chunks, we conducted the same experiment as in § 5 and find that probing significantly outperforms self-reported confidence in both accuracy and token efficiency. Figure 7 illustrates the results.

We also use prompting with designed verbalizer. Specifically, we append *"The answer is "* to each chunk, then obtain the probabilities of the next token being exactly *"correct"* or *"incorrect"*, and re-normalize them to produce a confidence score. However, the confidence obtained in this way is still less calibrated with the ECE score equal to 0.25. Using this forced self-reported confidence for early exit also performs much worse than our probe method in terms of both accuracy and token efficiency. Figure 8 illustrates the results.

## A.6 Further results

Table 10 and Table 11 show in-distribution probing performance measured by accuracy, precision, recall, and macro F1 across reasoning models and datasets.

Table 12 to Table 16 show out-of-distribution probing performance trained and test on representations from R1-Distill-Qwen-1.5B, R1-Distill-Qwen-7B, R1-Distill-Qwen-32B, and QwQ-32B, respectively.

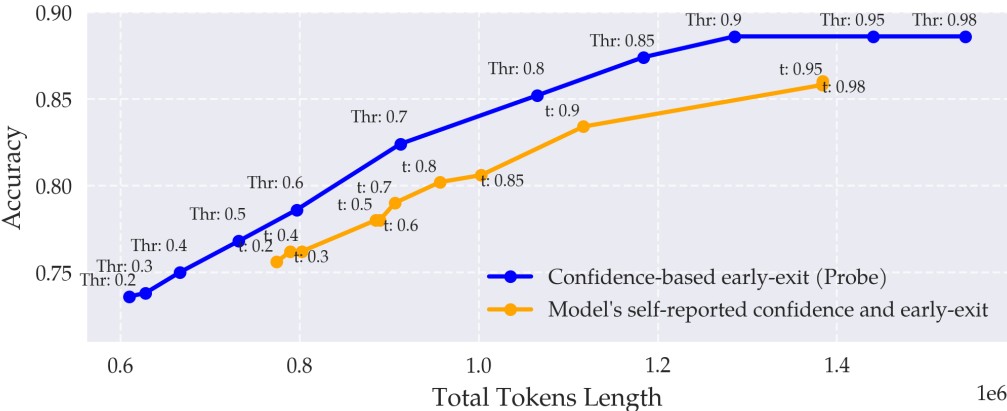

Figure 7: Final answer accuracy versus inference token cost with different early-exit strategies. For confidence-based early-exit, the curve is obtained by varying the probe confidence threshold (*Thr*) for answer correctness. For model's self-reported confidence and early-exit, the curve is generated by varying the model's self-reported confidence threshold (*t*).

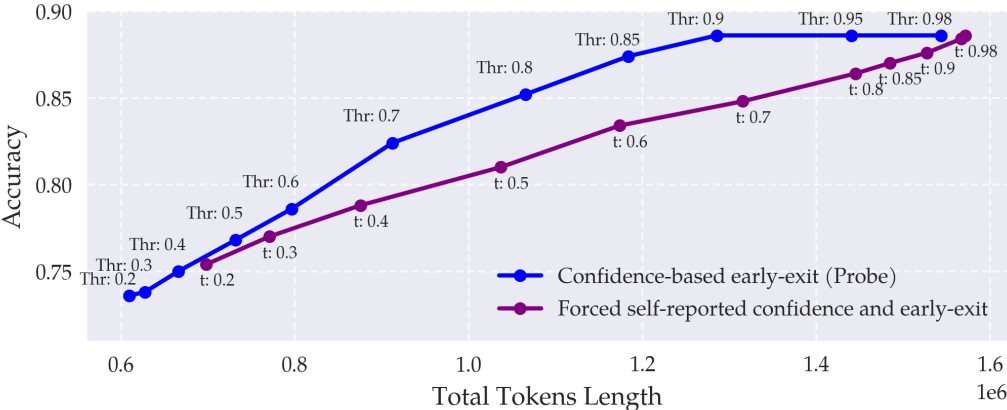

Figure 8: Final answer accuracy versus inference token cost with different early-exit strategies. For forced self-reported confidence and early-exit, we force the model to output confidence scores, and the curve is generated by varying the forced self-reported confidence threshold (*t*).

| Reasoning Model | GSM8K | | | | MATH | | | |
|---|---|---|---|---|---|---|---|---|
| | Accuracy | Precision | Recall | Macro F1 | Accuracy | Precision | Recall | Macro F1 |
| R1-Distill-Llama-8B | 0.77 | 0.85 | 0.82 | 0.73 | 0.80 | 0.84 | 0.88 | 0.75 |
| R1-Distill-Llama-70B | 0.91 | 0.92 | 0.97 | 0.82 | 0.89 | 0.92 | 0.93 | 0.83 |
| R1-Distill-Qwen-1.5B | 0.76 | 0.81 | 0.81 | 0.74 | 0.84 | 0.84 | 0.88 | 0.83 |
| R1-Distill-Qwen-7B | 0.84 | 0.88 | 0.92 | 0.77 | 0.87 | 0.89 | 0.94 | 0.82 |
| R1-Distill-Qwen-32B | 0.89 | 0.91 | 0.95 | 0.79 | 0.89 | 0.94 | 0.92 | 0.85 |
| QwQ-32B | 0.83 | 0.83 | 0.99 | 0.49 | 0.87 | 0.95 | 0.89 | 0.79 |

Table 10: Accuracy, precision, recall, and macro F1 score for probes trained and test on GSM8K and MATH datasets in in-distribution setting.

| Reasoning Model | AIME | | | | KnowLogic | | | |
|---|---|---|---|---|---|---|---|---|
| | Accuracy | Precision | Recall | Macro F1 | Accuracy | Precision | Recall | Macro F1 |
| R1-Distill-Llama-8B | 0.85 | 0.37 | 0.38 | 0.64 | 0.62 | 0.62 | 0.41 | 0.60 |
| R1-Distill-Llama-70B | 0.75 | 0.80 | 0.54 | 0.73 | 0.67 | 0.79 | 0.62 | 0.67 |
| R1-Distill-Qwen-1.5B | 0.83 | 0.45 | 0.62 | 0.71 | 0.72 | 0.23 | 0.53 | 0.42 |
| R1-Distill-Qwen-7B | 0.78 | 0.65 | 0.64 | 0.74 | 0.69 | 0.60 | 0.58 | 0.67 |
| R1-Distill-Qwen-32B | 0.91 | 0.88 | 0.96 | 0.91 | 0.70 | 0.80 | 0.67 | 0.69 |
| QwQ-32B | 0.82 | 0.84 | 0.85 | 0.82 | 0.78 | 0.82 | 0.88 | 0.74 |

Table 11: Accuracy, precision, recall, and macro F1 score for probes trained and test on AIME and KnowLogic datasets in in-distribution setting.

| Training Data | GSM8K | | MATH | | AIME | | KnowLogic | |
|---|---|---|---|---|---|---|---|---|
| | AUC ↑ | ECE ↓ | AUC ↑ | ECE ↓ | AUC ↑ | ECE ↓ | AUC ↑ | ECE ↓ |
| GSM8K | 0.82 | 0.04 | 0.90 (+0.06) | 0.07 (+0.04) | 0.75 (-0.05) | 0.14 (+0.04) | 0.62 (-0.05) | 0.08 (+0.01) |
| MATH | 0.82 (-0.01) | 0.10 (+0.06) | 0.84 | 0.03 | 0.84 (+0.04) | 0.18 (+0.08) | 0.63 (-0.04) | 0.14 (+0.08) |
| KnowLogic | 0.67 (-0.16) | 0.36 (+0.32) | 0.73 (-0.11) | 0.34 (+0.32) | 0.68 (-0.12) | 0.05 (-0.05) | 0.67 | 0.07 |

Table 12: ROC-AUC scores and ECE of trained probes on out-of-distribution test set. The numbers in red and green denote performance decrease and increase relative to the probe trained on in-distribution training set, respectively. R1-Distill-Qwen-1.5B is used as the reasoning model.

| Training Data | GSM8K | | MATH | | AIME | | KnowLogic | |
|---|---|---|---|---|---|---|---|---|
| | AUC ↑ | ECE ↓ | AUC ↑ | ECE ↓ | AUC ↑ | ECE ↓ | AUC ↑ | ECE ↓ |
| GSM8K | 0.82 | 0.04 | 0.86 (+0.02) | 0.06 (+0.03) | 0.76 (-0.04) | 0.15 (+0.05) | 0.60 (-0.07) | 0.17 (+0.10) |
| MATH | 0.86 (+0.04) | 0.06 (+0.02) | 0.84 | 0.03 | 0.73 (-0.07) | 0.18 (+0.08) | 0.68 (+0.02) | 0.17 (+0.10) |
| KnowLogic | 0.81 (-0.02) | 0.07 (+0.03) | 0.83 (-0.01) | 0.10 (+0.07) | 0.72 (-0.08) | 0.16 (+0.06) | 0.67 | 0.07 |

Table 13: ROC-AUC scores and ECE of trained probes on out-of-distribution test set. The numbers in red and green denote performance decrease and increase relative to the probe trained on in-distribution training set, respectively. R1-Distill-Qwen-7B is used as the reasoning model.

| Training Data | GSM8K | | MATH | | AIME | | KnowLogic | |
|---|---|---|---|---|---|---|---|---|
| | AUC ↑ | ECE ↓ | AUC ↑ | ECE ↓ | AUC ↑ | ECE ↓ | AUC ↑ | ECE ↓ |
| GSM8K | 0.82 | 0.04 | 0.87 (+0.03) | 0.04 (+0.01) | 0.98 (+0.17) | 0.17 (+0.07) | 0.73 (+0.06) | 0.06 (-0.01) |
| MATH | 0.89 (+0.06) | 0.03 (-0.01) | 0.84 | 0.03 | 0.97 (+0.16) | 0.10 (+0.00) | 0.72 (+0.05) | 0.15 (+0.08) |
| KnowLogic | 0.83 (+0.00) | 0.09 (+0.05) | 0.89 (+0.05) | 0.10 (+0.07) | 0.91 (+0.10) | 0.22 (+0.12) | 0.67 | 0.07 |

Table 14: ROC-AUC scores and ECE of trained probes on out-of-distribution test set. The numbers in red and green denote performance decrease and increase relative to the probe trained on in-distribution training set, respectively. R1-Distill-Qwen-32B is used as the reasoning model.

| Training Data | GSM8K | | MATH | | AIME | | KnowLogic | |
|---|---|---|---|---|---|---|---|---|
| | AUC ↑ | ECE ↓ | AUC ↑ | ECE ↓ | AUC ↑ | ECE ↓ | AUC ↑ | ECE ↓ |
| GSM8K | 0.82 | 0.04 | 0.88 (+0.04) | 0.09 (+0.06) | 0.71 (-0.09) | 0.17 (+0.07) | 0.62 (-0.04) | 0.25 (+0.18) |
| MATH | 0.87 (+0.05) | 0.06 (+0.02) | 0.84 | 0.03 | 0.75 (-0.05) | 0.16 (+0.06) | 0.73 (+0.06) | 0.20 (+0.13) |
| KnowLogic | 0.84 (+0.01) | 0.10 (+0.06) | 0.87 (+0.03) | 0.13 (+0.10) | 0.70 (-0.10) | 0.12 (+0.02) | 0.67 | 0.07 |

Table 15: ROC-AUC scores and ECE of trained probes on out-of-distribution test set. The numbers in red and green denote performance decrease and increase relative to the probe trained on in-distribution training set, respectively. R1-Distill-Llama-70B is used as the reasoning model.

| Training Data | GSM8K | | MATH | | AIME | | KnowLogic | |
|---|---|---|---|---|---|---|---|---|
| | AUC ↑ | ECE ↓ | AUC ↑ | ECE ↓ | AUC ↑ | ECE ↓ | AUC ↑ | ECE ↓ |
| GSM8K | 0.82 | 0.04 | 0.74 (-0.10) | 0.14 (+0.12) | 0.73 (-0.07) | 0.23 (+0.13) | 0.61 (-0.06) | 0.29 (+0.23) |
| MATH | 0.55 (-0.27) | 0.22 (+0.18) | 0.84 | 0.03 | 0.87 (+0.07) | 0.07 (-0.03) | 0.76 (+0.09) | 0.11 (+0.04) |
| KnowLogic | 0.61 (-0.22) | 0.14 (+0.11) | 0.81 (-0.03) | 0.05 (+0.02) | 0.84 (+0.04) | 0.07 (-0.03) | 0.67 | 0.07 |

Table 16: ROC-AUC scores and ECE of trained probes on out-of-distribution test set. The numbers in red and green denote performance decrease and increase relative to the probe trained on in-distribution training set, respectively. QwQ-32B is used as the reasoning model.

