# OpenReview forum: "Reasoning Models Know When They’re Right: Probing Hidden States for Self-Verification"
_colmweb.org/COLM/2025/Conference — COLM 2025_

### Official Review · Reviewer_5iSN · 2025-05-05

**Rating:** 6
**Confidence:** 4
**Ethics Flag:** 1

**Summary:**

The paper presents a method for on-policy control of reasoning language models. Specifically, the authors show that by using a simple (even linear) classifier on the hidden state it is possible to predict correctness of the intermediate answers during reasoning. The latter can be used to stop generation early without compromising the accuracy.
The finding is interesting and the authors provide experimental validation of their idea.

**Questions To Authors:**

1. Does the effect exist when using non-math datasets?
2. How often intermediate answers present in the reasoning traces? The statistic on the number of chunks / intermediate answers depending on the dataset would be interesting.
3. How sensitive is the reasoning data segmentation / captioning to the external LLM choice?

**Reasons To Accept:**

1. The author's method is simple and interesting for broad audience.
2. The experimental validation is sufficient to support the idea.
3. The language of the paper is clear.

**Reasons To Reject:**

1. While the idea of the work is interesting, the authors do not go anywhere beyond simple empirical observation of the effect.
No attempt was made to explain why the hidden states of reasoning LLMs predict correctness of intermediate answers. Also, it is hard to tell how novel is the idea comparing with the existing state of the art, as hidden states of LLMs were already used to detect hallucinations and to estimate answer uncertainty in previous works, for example in https://openreview.net/forum?id=LYx4w3CAgy. The authors should clearly explain the novelty of their approach.

2. Considering an empirical nature of the work, additional testing may be necessary. Specifically, the datasets used by the authors are mostly math-related. It is necessary to check the findings of the authors on non-math related datasets., for example NPHardEval.

3. The procedure for data preparation requires further validation. The authors claim that reasoning content contains multiple intermediate answers. This assumption is used to segment the reasoning trace into chunks with the help of an external LLM. Further, intermediate answers are extracted with an external LLM from each chunk. It would be nice to see some validation of the assumption that full intermediate answers exist in the reasoning data, since human reasoning usually does not contain complete intermediate answers. Also, the dependence of the chunking procedure on the external LLM should be studied.

---

> ### Author Response · Authors · 2025-06-02
>
> Thanks for the comments and feedback! We address concerns and provide more discussion below.
>
> **R1.**
>
> We thank the reviewer for these important points. We address both concerns below:
>
> **Regarding the explanation of why hidden states predict correctness:**
>
> Thanks for the question! We think it might have to do with data used to train reasoning models. Training with long CoT patterns requires the model to learn when it should backtrack, which may incentivize modeling the correctness of intermediate answers. This is somewhat validated in Section 4.4, where we show reasoning models have a better calibrated representation of correctness compared to non-reasoning models. We will add more explanation in Section 4.4. Overall we think how the correctness encoding emerges is an interesting question that is worth further investigation.
>
>
> **Regarding novelty compared to existing work:**
>
> We acknowledge that the probing method has already been widely used in existing works to explore model representation. However, we believe it’s worth redoing the experiments on reasoning models given their **unique search behavior** (See detailed discussion in ***General Response***). The probing results also reveal some interesting and new phenomena that have important implications for understanding reasoning models. Specifically:
> - Unique phenomenon: Our work addresses the "overthinking" phenomenon, which is **unique to reasoning models** (involving multiple reasoning paths via backtrackings) where they continue generating tokens after producing correct answers.
> - Different research question: Rather than detecting general hallucinations or uncertainty of a single response, we investigate whether reasoning models **know when they're right** during their reasoning process.
> - Model-specific insights: We demonstrate that reasoning models are **not using the correctness information optimally during inference, despite having information encoded implicitly.** This provides crucial insights for improving training and inference strategies specifically for reasoning models.
> - Remarkably low calibration error: It’s **interesting to see the calibration error is remarkably low** (especially compared with short CoTs generated by non-reasoning models in Section 4.4), which indicates the reasoning models inherently encode information about answer correctness and suggest that the self-verification ability is enhanced during long CoT supervised training.
>
>
> **R2 & Q1. More non-math dataset**
>
> Thanks for the suggestion. We already provide results on KnowLogic (see Figure 2, Table 1) in the paper, which is a **logical reasoning** dataset in multiple-choice form. Below we provide more results on GPQA (we are continuing running more models). They are consistent with prior results. The training data is GPQA_extended with the test set removed (altogether 348 examples), and the test set is GPQA_diamond. We will add the results to the paper.
> | Reasoning Model      | Accuracy | ROC-AUC | ECE   | Brier |
> |----------------------|----------|---------|-------|-------|
> | R1-Distill-Llama-8B  | 0.723    | 0.768   | 0.085 | 0.188 |
> | R1-Distill-Qwen-1.5B | 0.817    | 0.776   | 0.047 | 0.137 |
> | R1-Distill-Qwen-7B   | 0.751    | 0.765   | 0.045 | 0.178 |
> | R1-Distill-Qwen-32B   | 0.725    | 0.820   | 0.088 | 0.181 |
>
>
> **R3 & Q2. Statistics of intermediate answers**
>
> The existence of multiple intermediate answers is a unique **observation** from outputs of reasoning models. **Figure 1**, which is adapted from a real output from reasoning models, shows the long CoT contains multiple intermediate answers as the model keeps re-interpreting the user’s question. We have also provided the **detailed statistics in Table 4 in appendix** of the number of chunks (=number of intermediate answers) obtained after extracting each intermediate answer for each dataset and model combination. It can be seen that one output contains about 8 intermediate answers on average, and that it also correlates with reasoning model capability and data difficulty.
>
> **Q3. Sensitivity of LLM annotation**
>
> Thanks for your question.
> To test the sensitivity of LLM annotation, we compare annotation results from Gemini with GPT-4o-mini with randomly sampled 100 examples from MATH training data. As shown in the table below, the two models well agree with each other with 80% annotations exactly the same. The results show that the annotation task is relatively straightforward for state-of-the-art LLMs.
> | CoT Data      | Cohen's kappa coefficient | % consistent annotation | # segments |
> |---------------|---------------------------|-------------------------|------------|
> | Qwen1.5B+MATH | 0.58                      | 85                      | 3416       |
> | Qwen7B+MATH   | 0.62                      | 80                      | 1772       |
> | Qwen32B+MATH  | 0.60                      | 81                      | 1513       |

---

> > ### Author Response · Authors · 2025-06-06
> > **Thanks for your acknowledgement**
> >
> > Dear Reviewer 5iSN,
> >
> > Thank you for your acknowledgement of our response. If you have any remaining questions, please let us know. We are happy to address any remaining concerns before the end of the discussion period.

---

### Official Review · Reviewer_Etou · 2025-05-05

**Rating:** 7
**Confidence:** 4
**Ethics Flag:** 1

**Summary:**

This paper studies early exit for reasoning model, and finds that the hidden state of the intermediate answer can show the model's confidence to an answer based on the current generation. The authors then introduce a light verifier as a probe to enable the model exit reasoning step when the confidence is high. Results show that the probe could reduce output token by 24%, which accordingly reduces the model generation cost and latency.

**Questions To Authors:**

- The paper mentioned that using hidden states to evaluate confidence of non-reasoning model has a low performance. It could be more convincing to discuss the reason of such a difference.

**Reasons To Accept:**

- The idea of using hidden states to verify confidence is novel and clean;
- The writing is clear to understand and easy to follow;

**Reasons To Reject:**

- The paper could benefit from comparing prediction accuracy/efficiency with other methods. For example, for self-reported confidence, the model can generate a few tokens to estimate the current confidence at the end of each paragraph / chunk.
- Comparing to other reasoning tasks, math has a shorter and unique answer. This may reduce the difficulty to verify the answer confidence. The paper could benefit from testing the verifier accuracy on other tasks such as coding or formal proving.
- The authors mention that another language model is employed to extract the answer from a chunk. In real world deployment, using this method could be very costly and slow.

---

> ### Author Response · Authors · 2025-06-02
>
> Thanks for the constructive feedback! We provide more explanations and discussions below.
>
> **R1. Baseline of prompting for confidence verbalization**
>
> Thanks for the suggestion. Prompting the model to verbalize its confidence is a straightforward method to test its notion of correctness. However, in real implementation, we find it extremely difficult to get reasoning models to follow user instruction. Experimenting with different prompting strategies, we only manage to get the model to produce a verbalized confidence for 66% of the chunks. On these chunks, we conducted the same experiment as in Section 5 and find that probing significantly outperforms **self-reported confidence** in both accuracy and token efficiency. We provide additional figures to illustrate this result (see Figure 1):  [external materials](https://docs.google.com/document/d/1JHDijoJkkrm6oDhKWgLFczRU0vs2nQjidk8GqVOhDsY/edit?usp=sharing) .
>
> We also use prompting with designed verbalizer. Specifically, we append “The answer is ” to each chunk, then obtain the probabilities of the next token being exactly "correct" or "incorrect", and re-normalize them to produce a confidence score. However, the confidence obtained in this way is still less calibrated (ECE = 0.25). Using this **forced self-reported confidence** for early exit also performs much worse than our probe method in terms of both accuracy and token efficiency. See Figure 2 in [external materials](https://docs.google.com/document/d/1JHDijoJkkrm6oDhKWgLFczRU0vs2nQjidk8GqVOhDsY/edit?usp=sharing) .
>
> We will include these results in the paper.
>
>
> **R2. More non-math dataset**
>
> Thanks for the suggestion. We already provide results on KnowLogic (see Figure 2, Table 1) in the paper, which is a logical reasoning dataset in multiple-choice form. Below we provide more results on GPQA (we are continuing running more models). They are consistent with prior results. The training data is GPQA_extended with the test set removed (altogether 348 examples), and the test set is GPQA_diamond. We will add the results to the paper.
>
> As for coding problems, since there are hardly any intermediate answers during reasoning, they are not suitable for validating our findings.
> | Reasoning Model      | Accuracy | ROC-AUC | ECE   | Brier |
> |----------------------|----------|---------|-------|-------|
> | R1-Distill-Llama-8B  | 0.723    | 0.768   | 0.085 | 0.188 |
> | R1-Distill-Qwen-1.5B | 0.817    | 0.776   | 0.047 | 0.137 |
> | R1-Distill-Qwen-7B   | 0.751    | 0.765   | 0.045 | 0.178 |
> | R1-Distill-Qwen-32B   | 0.725    | 0.820   | 0.088 | 0.181 |
>
>
> **R3**
>
> Thanks for your comments. Our main focus is to explore whether reasoning models encode correctness information. The finding points to potential inference-time intervention to improve efficiency, but we do acknowledge that the method in the paper is more of a post-hoc analysis and a more efficient online answer extractor would be needed if deployed. Thanks for pointing that out. We will add more discussion about it in the discussion and limitation section.
>
>
> **R4**
>
> Thanks for the question! The major decrease in performance of non-reasoning models comes from the higher calibration error. We think it might have to do with training data difference. Non-reasoning models are trained with question-response pairs that contain a single correct answer, whereas reasoning models usually undergo long CoT supervision which contains intermediate answers that can be right or wrong. Training with long CoT patterns might lead models to implicitly learn when it should backtrack, which largely corresponds to erroneous intermediate answers.

---

> > ### Author Response · Authors · 2025-06-06
> > **Looking forward to your reply**
> >
> > Dear Reviewer Etou,
> >
> > Thanks for your comments. We believe our response have addressed your concerns above. We would greatly appreciate it if you could take a look at our response and let us know if you have any remaining questions. We look forward to addressing any remaining concerns before the end of the discussion period.

---

> > ### Comment · Reviewer_Etou · 2025-06-08
> > **Response to the rebuttal**
> >
> > Hi authors,
> >
> > Thank you for your response to my concerns. Below is my feedback:
> >
> > ---
> >
> > > R1 baseline methods
> >
> > The additional baseline results clearly address my concerns.
> >
> > > R2 non-math dataset
> >
> > The additional results on GPQA is a strong support to the hidden verifier method. However, I still have the concern on the method's generalizability, since the hidden verifier cannot handle tasks without intermediate answers, while many tasks have the same feature, and it is even worse that identifying whether the task has intermediate answer or not is also challenging and costly.
> >
> > > R4 underlying mechanism for the findings when using hidden verifiers for non-reasoning model
> >
> > It will be more convincing to have an experiment on the author's hypothesis. However, I'd acknowledge that this question might be out of the scope of this paper.
> >
> > ---
> >
> > I've accordingly modified the score since the rebuttal addresses some of my concerns.

---

> > > ### Author Response · Authors · 2025-06-09
> > > **Response to Reviewer Etou**
> > >
> > > Thanks for your kind reply! We are happy that our response has addressed most of your concerns. We will add more discussion on the limitation of the method and potential future work to the paper. Thanks again for the duscussion!

---

### Official Review · Reviewer_KxZk · 2025-05-11

**Rating:** 6
**Confidence:** 4
**Ethics Flag:** 1

**Summary:**

This paper studies the role of the hidden state in reasoning models in verifying the correctness of their intermediate reasoning process.
They find that using a simple layer of MLP can probe the correctness from the hidden states and can generalize to OOD data.
They conduct an early-exit experiment on MATH dataset.

**Questions To Authors:**

1. In Sec 4.4, the encoding of correctness is related to long CoT reasoning abilities. Does this suggest that using internal hidden states does not generalize to smaller/weaker models?

**Reasons To Accept:**

1. This paper shows that the hidden state in reasoning models can serve as logit lens with a single MLP.
2. The experimental results show that the token usage can be reduced by 24% given same reasoning accuracy.

**Reasons To Reject:**

1. There are existing works interpreting the hidden states in the reasoning process of the LLMs[1]. I think the differences between RMs and LLMs are not big enough to review the role of the hidden states again. I would suggest a more thorough investigation into logit lens employed in LLMs.
2. Lack of compared baselines. There are lots of work focusing on either external verifiers or internal verifiers. For example, [2] exploits an external verifier for enhancing reasoning performance and [3] utilizes hidden states for internal consistency. At least both of them can be compared to your method regarding accuracy and token efficiency.

[1] Li, Zhaoyi, et al. "Understanding and patching compositional reasoning in llms." ACL 2024.

[2] Zhu, Tinghui, et al. "Deductive beam search: Decoding deducible rationale for chain-of-thought reasoning." COLM 2024.

[3] Xie, Zhihui, et al. "Calibrating reasoning in language models with internal consistency." NeurIPS 2024.

---

> ### Author Response · Authors · 2025-06-02
>
> We thank the reviewer for the comments. We would like to firstly emphasize the distinctions that necessitate our investigation. Existing work that interprets hidden states of short CoTs cannot address our central research question: **whether reasoning models know when they're right** during backtracking, especially **given the existing "overthinking" phenomenon**. Our motivation distinguishes our work from previous research that mainly focuses on understanding and improving non-reasoning models [1,2,3]. Please refer to our ***General Response*** for detailed motivation and novelty discussion.
>
>
> **R1. Why it’s worth revisiting reasoning models’ representations**
>
> Our research question focuses on whether reasoning models know when they're right. The answer to this question is not obvious given the observed “overthinking” phenomenon and the reported discrepancy in model’s inner representations and behavior. The question also has far more significance given that reasoning models are **uniquely characterized by search behavior during reasoning, and such representation of correctness becomes crucial to the success of reasoning**. Therefore, we believe it is **important to revisit hidden states of reasoning models**.  Our results also show that reasoning models exhibit substantially better calibration, which suggest that further research is essential for understanding what and how reasoning models learn during training. Compared to [1] that adopts layerwise logit lens analysis, **the simplicity and proven success of our probing method further proves the importance of posing reasoning models as a unique research target**.
>
>
> **R2. Comparison with other verifier/reasoning efficiency works**
>
> Thanks for the pointer. We would like to clarify that our primary motivation is to explore a fundamental scientific question rather than propose an efficient inference method or verifier. Our work focuses on understanding **whether reasoning models know when they're right**.
>
> [2] trains an external verifier that validates the correctness of current reasoning and uses the verifier to assist in beam search decoding to mimic search behavior for non-reasoning models. [3] uses layer-wise decoding consistency to measure model confidence in current answer and boosts performance by re-weighting ensemble answers in inference time. Both [2] and [3] focus on non-reasoning models and far more effort is required to validate answer correctness compared to our probing method. This also indicates the difference between reasoning and non-reasoning models, where reasoning models that naturally exhibit search behavior, and have a better representation of answer correctness that can be easily extracted. We will add more discussion on how our method relates to previous works on non-reasoning models in the paper.
>
>
> **Q1.**
>
> Thanks for the question! The critical factor is not model size per se, but whether the model has undergone reasoning-specific training. As we demonstrate in Section 4.4, reasoning models exhibit significantly better-calibrated notions of correctness compared to their non-reasoning counterparts. We think it might have to do with training data difference. Non-reasoning models are trained with question-response pairs that contain a single correct answer, whereas reasoning models usually undergo long CoT supervision which contains intermediate answers that can be right or wrong. This exposure appears to enable reasoning models to implicitly learn rich information about the reasoning process, even without explicit supervised signals for correctness encoding. As shown in Figure 2 and Table 1, even smaller reasoning models like DS-Distill-Qwen-1.5B show remarkably low calibration error along with high accuracy.

---

> > ### Author Response · Authors · 2025-06-06
> > **Further response to Reviewer KxZk**
> >
> > We acknowledge that there are works that study how to reason more efficiently for CoT-prompted non-reasoning models. While most of them are built upon best-of-n sampling based on non-reasoning models, reasoning models already perform sequential test-time scaling by search. Such discrepancy makes comparison less straightforward. The step-wise beam search in [2] requires reasoning to be generated step by step, while reasoning models cannot be easily prompted to enforce step-by-step generation as in [2]. The internal consistency score used in [3] relies on logits for unambiguous single-token answers (e.g., “True” or “False”, or “A/B/C/D”), whereas the complex reasoning tasks (e.g., math problems) do not fit into the format.
> >
> > Nevertheless, we adopt the trained deductive verifier model released in [2] for evaluating intermediate answers in our setting. We treat each intermediate answer and their corresponding reasoning as “the next step” for evaluation and the results are shown below and in Figure 3 in [external materials](https://docs.google.com/document/d/1JHDijoJkkrm6oDhKWgLFczRU0vs2nQjidk8GqVOhDsY/edit?usp=sharing). It can be seen that our lightly trained probe outperforms the deductive verifier in both discrimination performance and improving reasoning efficiency (applied to early exit).
> >
> > | Model                                   | Accuracy | ROC-AUC | Macro F1 | ECE  | Brier |
> > |-----------------------------------------|----------|---------|----------|------|-------|
> > | Verifier from deductive-beam-search [2] | 0.70     | 0.53    | 0.48     | 0.09 | 0.22  |
> > | Our probe                               | **0.80**     | **0.84**    | **0.75**     | **0.03** | **0.14**  |

---

> > > ### Author Response · Authors · 2025-06-06
> > > **Looking forward to your reply**
> > >
> > > Dear Reviewer KxZk,
> > >
> > > Thanks for your time in reviewing. Since the discussion period has started, we would greatly appreciate it if you could take a look at our response and let us know if you have any remaining questions. We look forward to addressing any remaining concerns before the end of the discussion period.

---

> > ### Comment · Reviewer_KxZk · 2025-06-09
> >
> > I think the authors have convinced me of the differences between LLMs and RMs and the comparison between the baselines and their method has proven the effectiveness. So I will raise my score.

---

> > > ### Author Response · Authors · 2025-06-09
> > > **Response to Reviewer KxZk**
> > >
> > > Thanks for your reply! We will add more discussion on distinction between LLMs and RMs and the further experimental results to our paper.

---

### Author Response · Authors · 2025-06-02
**General response**

We thank all reviewers for their constructive comments and suggestions. We would like to make a few important clarifications on the motivation and novelty of our work.

**Motivation and Main Focus**

The question we want to study is **whether reasoning models represent correctness of intermediate answers during reasoning**, which is crucial to the success of the unique search behavior (characterized by backtracking to previous steps during reasoning) **exclusively** present in reasoning models.

We note that the answer to the question is not obvious. Despite the overall success in reasoning effectiveness, the observed overthinking phenomenon seems to suggest the other way around (discussed in the **Introduction Section**). Existing works that report discrepancy [1] in model behavior and inner state information and the revealed information in model hidden representation [2] further blur the conclusion. Our results also show that these models are distinct from non-reasoning models—exhibiting substantially better calibration—and suggest that further research is essential for understanding what and how reasoning models learn during training.

**Difference from Prior Results**

As stated above, the motivation stems from **search behavior and observed “overthinking”, which is unique to reasoning models**.

We also demonstrate in Section 4.4 that reasoning models exhibit significantly better-calibrated notions of correctness than their non-reasoning counterparts. The universally low calibration error along with high accuracy (even on smaller models like DS-Distill-Qwen-1.5B) is remarkable compared to previous works that probe non-reasoning models' hidden states to test models’ ability to know its own knowledge boundary [2] or detect hallucination [3]. The technique we adopt is also much simpler than prior works on non-reasoning models’ representation [4,5,6]. This observation further **underscores the uniqueness of reasoning models and reinforces the necessity for dedicated investigation into their distinctive properties**.


[1] Orgad, H., Toker, M., Gekhman, Z., Reichart, R., Szpektor, I., Kotek, H., & Belinkov, Y. (2024). Llms know more than they show: On the intrinsic representation of llm hallucinations. arXiv preprint arXiv:2410.02707.

[2] Kadavath, S., Conerly, T., Askell, A., Henighan, T., Drain, D., Perez, E., ... & Kaplan, J. (2022). Language models (mostly) know what they know. arXiv preprint arXiv:2207.05221.

[3] Sriramanan, G., Bharti, S., Sadasivan, V. S., Saha, S., Kattakinda, P., & Feizi, S. (2024). Llm-check: Investigating detection of hallucinations in large language models. Advances in Neural Information Processing Systems, 37, 34188-34216.

[4] Li, Zhaoyi, et al. "Understanding and patching compositional reasoning in llms." ACL 2024.

[5] Zhu, Tinghui, et al. "Deductive beam search: Decoding deducible rationale for chain-of-thought reasoning." COLM 2024.

[6] Xie, Zhihui, et al. "Calibrating reasoning in language models with internal consistency." NeurIPS 2024.

---

### Decision · Program_Chairs · 2025-07-08

**Decision:**

Accept

**Comment:**

Overall quality: This paper presents an empirical study on leveraging hidden states in reasoning language models to enable early-exit generation through a lightweight MLP-based verifier. The empirical setup is sound, and the findings are clearly demonstrated, with a promising application for reducing generation cost and latency for math reasoning tasks. However, the scope of experimentation is somewhat narrow (mostly focused on math reasoning), and the novelty relative to existing literature is a bit unclear. As a result, reviewers consistently point out the need for more baselines, more datasets, and clearer positioning relative to prior work on internal verifiers and logit lens methods.

Presentation clarity: The paper is clearly written and easy to follow. All three reviewers appreciated the presentation quality. The methodology and results are presented with sufficient detail for replication.

Originality: While the application of probing hidden states with a simple verifier is not entirely new, the specific investigation on reasoning models is relatively underexplored.

Significance: The idea has practical implications for making reasoning models more efficient, especially in scenarios where latency or token cost is critical. However, the current evaluation is mostly limited to math problems, and thus the generality and robustness of the approach remain unclear. Its broader significance would be enhanced by extending the study to other domains like coding, theorem proving and more.

Pros:
- Clear and well-written presentation.
- Simple and effective method for early exit in reasoning models.
- Demonstrates tangible gains in token efficiency (24% reduction).
- Offers a practical, lightweight solution applicable to existing models.

Cons:
- Unclear position compared to existing probing methods on LLMs, thus raising concerns about the limited comparison with relevant baselines from recent literature.
- Evaluations are confined to math reasoning tasks, limiting generalizability.
- Relies on external LLMs for data segmentation and annotation, raising scalability and robustness concerns.
- Lack of deeper analysis on why hidden states are predictive of reasoning correctness.